# Dissatisfaction with Body Weight among Polish Adolescents Is Related to Unhealthy Dietary Behaviors

**DOI:** 10.3390/nu12092658

**Published:** 2020-08-31

**Authors:** Agata Wawrzyniak, Joanna Myszkowska-Ryciak, Anna Harton, Ewa Lange, Wacław Laskowski, Jadwiga Hamulka, Danuta Gajewska

**Affiliations:** 1Department of Human Nutrition, Institute of Human Nutrition Sciences, Warsaw University of Life Sciences (WULS), 159C Nowoursynowska Str, 02-776 Warsaw, Poland; jadwiga_hamulka@sggw.edu.pl; 2Department of Dietetics, Institute of Human Nutrition Sciences, Warsaw University of Life Sciences (WULS), 159C Nowoursynowska Str, 02-776 Warsaw, Poland; joanna_myszkowska_ryciak@sggw.edu.pl (J.M.-R.); anna_harton@sggw.edu.pl (A.H.); ewa_lange@sggw.edu.pl (E.L.); 3Department of Organization and Consumption Economics, Institute of Human Nutrition Sciences, Warsaw University of Life Sciences (WULS), 159C Nowoursynowska Str, 02-776 Warsaw, Poland; waclaw_laskowski@sggw.edu.pl

**Keywords:** dissatisfaction with body weight, socio-demographic factors, dietary behaviors, adolescents

## Abstract

The aims of the study were to determine the socio-demographic factors that may affect body weight dissatisfaction and to analyze the relationship between eating habits and dissatisfaction with body weight among a national random sample of Polish adolescents aged 13–19 years. Data on gender, age, level of education, body weight status, screen time, body weight satisfaction and selected nutritional behaviors were collected using a questionnaire. Body mass status was assessed based on weight and height measurements. A total of 14,044 students from 207 schools participated in the study. A significant effect of gender, age, level of education, body weight status and screen time status on the participants’ dissatisfaction with the body weight was observed. The greater prevalence of body weight satisfaction was observed among boys, younger subjects, secondary school students, adolescents with normal body weight status and those with screen time up to 2 h. Whereas girls, older study participants (17–19 years old), overweight/obese adolescents and subjects with screen time over 4 h were more often dissatisfied with body weight. Furthermore, it has been shown that participants dissatisfied with their body weight less often met dietary recommendations. These findings can help dietitians, nutritionists and healthcare professionals to provide age-specific and gender-specific nutrition strategies to promote healthy lifestyle among school-going adolescents.

## 1. Introduction

The process of maturation, called adolescence, depends on genetic, sexual, environmental and cultural factors. This process is characterized by transformations in the body structure and in the physical appearance (biological maturation), psyche (mental maturation), attitude towards sex (psychosexual maturation), in fulfilling the social roles (social maturation). All these factors affect the perception of one’s body and body satisfaction [1,2]. During adolescence, eating habits related to body weight, body weight dissatisfaction and health also develop intensively [3]. Nutritional expertise in this area and the determination of the relationship between dissatisfaction with body weight and non-compliance with healthy eating recommendations are becoming increasingly prominent. In fact, an unhealthy diet during the period of intense growth may affect the development of adolescents, trigger the non-acceptance of the body as a consequence, and may cause diet-related diseases in adulthood [2,4,5,6]. Currently, the mass media promote a model of slim figure in women and muscular/athletic in men, advertising food products and diets with inadequate nutritional value, which can also lead to undesirable nutritional and health behaviors and non-acceptance of body weight [5,6,7,8].

In light of the above circumstances, nutrition strategies aimed at maintaining the health of children and adolescents are fundamental and their development is extremely important for most countries in the world especially in groups at risk of improper nutrition. Additionally, the World Health Organization (WHO) highlighted the health promotion and the reduction in health inequalities as priorities for Europe in the Health 2020 program. However, in order to establish specific goals to maintain health in each country, it is necessary to define target groups according to age, gender and level of education [5].

In Poland, many studies concerning irregularities in body weight and eating behaviors of adolescents have been carried out [9,10,11,12,13,14]. However, these studies were provided on rather small population, usually at a selected age or covered only selected nutritional behaviors without the study of dissatisfaction with body weight [9,10,11,12,13]. Few studies concerned the methods of body weight regulation used by adolescents [11,12] or body weight dissatisfaction but with a small number of analyzed food products [14]. The only study involving a large representative group of Polish adolescents is the Health Behavior of School-age Children (HBSC) [5,6], conducted for a period of over 30 years across more than 40 countries, including Poland. However, the HBSC study presents the self-assessment of body weight by Polish adolescents and only in groups of 11-, 13- and 15-year-old boys and girls, and does not examine the relationship between self-assessment of body weight and nutritional behaviors. Thus, our study on dissatisfaction with the body weight of Polish adolescents related to unhealthy eating behavior is unique, especially in Poland, and conducted among a large and representative population, aged 13–19 years old (*n* = 14,044). To our knowledge there is no nationwide research covering all age categories of adolescents dissatisfied with body weight in respect to compliance with nutritional recommendations. This study was undertaken to examine: (1) how socio-demographic factors determine dissatisfaction with body weight (age, gender, body weight status, screen time); (2) how dissatisfaction with body weight influences the nutritional behavior of adolescents aged 13–19 years. The obtained results can be helpful in targeted and effective nutrition education in selected groups according to age, gender, level of education, to improve the quality of the diet and the health of young people in the future.

## 2. Materials and Methods 

### 2.1. General Information

This study is part of the research and education program Wise Nutrition-Healthy Generation (WNHG, Warsaw, Poland), awarded by The Coca-Cola Foundation and addressed as free of charge to secondary and high schools, students, their parents and teachers. The program was conducted within years 2013–2015 by the Polish Society of Dietetics, among 2058 schools (attended by almost 450,000 students). The main objective of the program was to educate the secondary and upper secondary school students about the role of healthy eating and physical activity in the prevention of diet-related diseases. The research part of the program included the evaluation of selected eating behavior and the assessment of body weight status among Polish adolescents aged 13–19 years. A detailed description of the program as well as the research methodology has been described in previously published articles on nutritional behaviors [15], body weight status [16] and screen time among Polish adolescents [17].

The program was carried out following the requirements by the Helsinki Declaration. The protocol of the study was approved by the Scientific Committee of the Polish Society of Dietetics (PTD-ZG-1/05/2013). The program received the patronage of the Minister of National Education and local government educational institutions. All parents were asked for their written consent to the participation of students in the study. In addition, students over 16 years of age also had to provide their written consent. The personal data of participants were fully anonymized. 

This paper focused on the relationship between satisfaction with body weight and nutritional behaviors among Polish adolescents, as well as on the factors that affect body dissatisfaction.

### 2.2. Study Participants

The representative sample of Polish adolescents were recruited from schools participating in the project. A detailed method of recruiting schools and subjects has been described elsewhere [15,16,17]. In brief, schools were randomly selected using the stratified sampling method from the total of 2058 enrolled institutions. Schools were categorized by province (based on administrative division), size of the place of residence (village, small town, medium city, city), as well as the level of education (secondary for all adolescents aged 13–16 years and upper secondary for youth from 16 to 20 years of age). In the selected schools, students were randomly drawn from the class register. Exclusion criteria were diagnosed disease that require a special diet, pregnancy or lactation in girls or lack of written consent. Based on the adopted criteria, 207 schools were selected for the study (~10% of the total number of 2058 schools enrolled in the project) and 14,044 students, including 7553 girls and 6491 boys aged 13–19 years old.

### 2.3. Socio-Demografic Data and Eating Behavior Survey

Socio-demographic data (gender, age, level of education, i.e., secondary school 7–9 years of education, upper secondary school 10–13 years of education, screen time, i.e., time spent in front of the screen in a passive way during a typical day), data on body weight satisfaction (the question “Are you satisfied with your body weight?” with possible “yes” or “no” answers) and data on selected students’ eating behavior were collected by qualified dietitians (undergraduate or graduate studies) using the Pen-and-Paper Personal Interview (PAPI) method. The questionnaire and anthropometric measurements were carried out prior to nutritional education. The research was conducted simultaneously in Poland. According to the self-reported satisfaction with body weight all respondents were divided into two groups: adolescents satisfied with their body weight (SBW) and adolescents dissatisfied with their body weight (DBW). 

The nutritional part of the questionnaire consisted of nine questions with possible “yes” or “no” answers. The first six questions were related to positive eating behavior according to the Polish Pyramid of Healthy Nutrition and Physical Activity [18]: (1) eating breakfast every day before going to school; (2) daily consumption of at least one portion of fresh fruit; (3) daily consumption of at least two portions of vegetables; (4) daily consumption of milk and/or milk fermented beverages; (5) daily consumption of whole grains; (6) eating fish at least once a week. The last three questions concerned unfavorable nutritional practices, such as: (1) drinking sweetened non-alcoholic beverages (carbonated drinks and other non-alcoholic non-carbonated drinks) several times a week; (2) eating sweets more than once a day; (3) eating fast food more than twice a week. The questionnaire was validated prior to the main study [15,16,17]. All the nutritional behaviors of the respondents were analyzed based on the division into SBW and DBW groups.

### 2.4. Anthropometric Measurements

To determine body weight status, anthropometric measurements of the body weight and height of participants were performed by trained dietitians in accordance with a standard procedure developed by the National Health and Nutrition Examination Survey (NHANES) [19]. Weight measurements were carried out using a digital floor scale (TANITA HD-380 BK, Tanita Corporation, Tokyo, Japan); steel strips (0–200 cm) were used to measure the height. A detailed description of all procedures was provided in previously published articles [15,16,17]. Body mass index (BMI) was calculated as body weight in kilograms divided by the square of height in meters. For students aged 18 and younger, calculated BMI value was plotted on gender BMI centile charts for age (with an accuracy of one month) [20] and the percentile value was obtained from percentile grids. As recommended by the International Obesity Task Force (IOTF) cut-offs, the sample was classified into four body mass categories according to percentiles: below the 5th percentile as underweight, between the 5th and 84th as normal weight, between the 85th and 94th as overweight and the 95th percentile and above as obese [21,22]. The WHO standard BMI criteria were applied for individuals above the age of 18 years old as follows: BMI < 18.5 kg/m^2^—underweight; BMI 18.5–24.9 kg/m^2^—normal body weight; BMI 25–29.9 kg/m^2^—overweight; BMI ≥ 30 kg/m^2^—obesity [22].

### 2.5. Statistical Analysis

Statistical analysis was carried out using the IBM SPSS Statistics25 (SPSS Inc., Chicago, IL, USA) software package. The results were presented as percentages according to the nominal variable. Data were analyzed in the groups of students satisfied (SBW) and dissatisfied (DBW) with their body weight in terms of age, gender, level of education, body weight status and screen time. Statistical significances for nominal variables were determined using the Pearson’s chi-square test as a test of independence. The correspondence analysis was performed and odds ratio (the Wald test) with 95% confidence intervals (95% CI) was calculated using logistic regression analysis to study the relationship between body weight dissatisfaction and eating behaviors. The value of *α* = 0.05 was considered as statistically significant.

## 3. Results

A total of 14,044 teenagers aged 13 to 19 years participated in the study. The characteristics of the group according to gender, age, level of education, body weight status and screen time are presented in Table 1. A significant relationship of gender, age, level of education, body weight status and screen time on the participants’ dissatisfaction with the body weight was observed.

The greater prevalence of body weight satisfaction was observed among boys, younger subjects (i.e., 13–14 years), secondary school students, adolescents with normal body weight status and those with screen time up to 2 h. Whereas girls, older study participants (17–19 years), overweight and obese adolescents, subjects with screen time over 4 h were more often dissatisfied with body weight. Among DBW adolescents, significantly fewer individuals had normal body weight, more adolescents were overweight or obese (with a 2-fold higher odds ratio for overweight and a 5-fold higher odds ratio for obesity).

After adopting the SBW group as the reference category, a significantly different odds ratio for the analyzed eating behavior was observed especially for having breakfast every day before leaving for school, consuming vegetables every day (at least two servings), drinking milk or fermented milk beverages every day, consuming fish at least once a week, consuming sweets more than once a day (Table 2). Students from the DBW group less often met dietary recommendations (except for eating sweets). No statistically significant odds ratio was observed between the SBM and DBW groups and eating behavior in regards to: consuming fresh fruit every day (at least one serving), consuming whole-grained bread every day, drinking sweet beverages few times a week and consuming fast food more than two times a week.

After adopting participants with normal body weight status as the reference category, a significantly different ORs for the consumption of some food products were noted among DBW students, compared to SBW students, especially for sweets, fast food or drinking sweet beverages. In the DBW group, underweight students characterized a higher OR value for the consumption of fast food, sweets and sweet beverages, while in the case of overweight and obese students, the OR values for the consumption of fast food, sweets and sweet beverages were lower compared to students with a normal body weight status in this group. The inverse relationship was observed in the evaluation of fruit consumption. It should be emphasized that in the case of so-called healthy food products (vegetables, whole-grained bread, milk or milk beverages, fish) OR values did not differ so significantly among DBW students.

Figure 1 presents the relationship between the satisfaction or dissatisfaction with body weight in students and selected eating behaviors in the total group. The correspondence analysis revealed that beneficial nutritional behaviors, such as consuming breakfast, fruit, vegetables, whole-grain bread, milk or milk beverages and fish, were linked together. Whereas unfavorable eating behaviors, such as skipping breakfast, low consumption of milk products, fruits, vegetables, fish and whole-grain bread consumption, were related to each other. Compliance with body weight satisfaction corresponded to the beneficial eating behaviors. Dissatisfaction with body weight was associated with adverse nutritional behaviors. Eating sweets and eating fast food as well as drinking sweet beverages were not directly associated with healthy eating in any of the groups.

## 4. Discussion

The present study confirmed that socio-demographic factors such as gender, age, body weight status, level of education, as well as screen time are associated with body weight satisfaction. Furthermore, adolescents dissatisfied with their body weight declared unhealthy dietary behavior more often.

### 4.1. Dissatisfaction with Body Weight and Socio-Demographic Factors

In our study, body dissatisfaction was more often recorded in female adolescents and increased in both genders with age of the respondents. Several international studies have shown that a percentage of girls dissatisfied with their appearance increases with age, although studies provided by Ames et al. [23], Carter et al. [24], Jankauskiene, Baceviciene [2] did not confirm overweight and obesity in most of them. The body weight overestimation was more common in girls, whereas the underestimation was more common in boys [2]. The high prevalence of body weight dissatisfaction among female adolescents is define in the literature as “normative discontent” [25]. During adolescence, the body weight satisfaction is very important as it affects life satisfaction and proper social interaction among peers [6,26,27]. Adolescents with an incorrect body mass index and those who underestimate or overestimate their body weight reported a higher body dissatisfaction, social physique anxiety, disordered eating and lower self-esteem [2,12]. Adolescents, however, have difficulties with an adequate assessment of their body weight, and often overestimate or underestimate their body weight. Studies showed that the body weight perception depended on the age and gender of respondents [2,6,28,29]. In a representative HBSC study, conducted in Poland in 2018 among 5225 students aged 11–15 years, only half of adolescents with normal body weight were able to properly assess their body weight. The boys have evaluated it correctly more often than the girls. In the general group of teenagers with normal body weight, over one-third rated their body weight as too high and one in seven adolescents as too low. The percentage of adolescents who rated their body weight as low (mainly because of boys’ response) and those who rated their body weight as too high (mainly because of girls’ response) increased with age [6]. 

As documented in earlier study [17], the average screen time increased with age, which can also cause an increase in dissatisfaction with body weight. Technological progress, digitization of all areas of life, as well as the increasing availability of electronic equipment, including computers, smartphones, televisions, affect children and young people, promote an attractive figure, as well as a sedentary lifestyle [7,8,12,17]. The results of subsequent rounds of international HBSC studies also indicated that Poland is a country where girls had the least favorable and more disturbed assessment of their body weight [30]. Polish, 15-year-old girls considered themselves too fat more often than their peers from 42 other countries, although this was not confirmed in BMI categories [6,31]. 

### 4.2. Dissatisfaction with Body Weight and Compliance with Nutritional Recommendations

Perceiving one’s body weight as deficient or excessive can become a source of various psycho-emotional problems and nutrition irregularities [32,33,34]. The pyramid of healthy nutrition and lifestyle of children and adolescents, as well as the principles of healthy eating published in Poland in 2019 [35] recommend, among others: eating five meals regularly (including breakfast), eating various vegetables and fruits as often as possible and as much as possible, drinking at least three to four glasses of milk a day (you can replace them with plain yogurt, kefir and—partly—cheese), eating cereal products, especially whole grains, eating lean fish (at least once a week), avoiding sweetened beverages and sweets (replacing them with fruits and nuts), avoiding salty snacks and fast food. This study showed that students who were dissatisfied with body weight were less likely to make healthy nutritional choices.

Numerous studies provided in different countries showed that skipping breakfast and lower meal frequency were more common in adolescents with higher BMI or body weight dissatisfaction [2,12,15,36,37,38]. It can be hypothesized that skipping breakfast can be a strategy for losing weight for teenagers. However, eating breakfast every day has proved to have a beneficial effect on the development, health and functioning of young people at school, including: reducing the risk of developing overweight and obesity [39,40], by consuming the essential nutrients and reducing the consumption of snacks and high-energy products that can be harmful to health [41]. In addition, it was proved that students who did not eat breakfast more often showed other unfavorable eating behavior (low consumption of fruit, vegetables, fish and whole wheat bread), as these products are often included in a typical Polish breakfast [15]. Thus, our results indicate a strong need for the increase in educational activities promoting regular consumption of breakfast, especially among adolescents with underweight or overweight/obesity, or with body weight dissatisfaction.

Proper eating habits are very important for the development of a young body, but changes in eating habits often do not comply with the recommendations. In practice, the implementation of proper nutrition recommendations in these population groups is very difficult due to the existing barriers, e.g., availability of healthy food, inadequate nutritional knowledge of caregivers and teenagers and personal food preferences [6,42,43,44]. Qualitative and quantitative abnormalities in nutrition contribute to underweight or overweight/obesity and/or dissatisfaction with body weight in children and adolescents, which increases the risk of some diet-related diseases. Adolescents dissatisfied with body weight usually change eating habits and limit so-called health products in diet (fruit and vegetables, dairy products, whole grain bread, fish), but first of all modify the consumption of unfavorable food (fast food, sweets, sweet beverages), as a source of sugar and fat. Students with underweight or overweight/obesity and dissatisfied with their body weight were more likely to reduce (overweight, obese, with overestimated body weight) or increase (underweight, with underestimated body weight) the consumption of fats, spreads and oils, sweets, sweet drinks, fast foods, i.e., unhealthy foods in their diet, than they consumed healthy products, as recommended, i.e., vegetables, dairy products, whole grains, and fish [2,14,36]. A study conducted among girls in Lithuania [2], showed that a higher BMI and/or body weight overestimation were associated with having less sweets and consuming less fats, spreads, and oils. In boys, the BMI was associated with consuming less fruits and berries. The boys’ body weight overestimation was related to a lower consumption of milk, cheese, yogurt, fats, spreads, and oils. Adolescents with a higher BMI and body weight overestimation demonstrated higher body image concerns, and a poorer eating-related behavioral profile. In studies conducted in Slovakia [28] among adolescents aged 13 to 15 who assessed their body weight as too high, the most frequent weight-reduction behaviors were drinking more water, eating more fruits and vegetables, and consuming fewer sweets and soft drinks. Girls prevailed above boys in the use of these dietary methods. In Zarychta et al. [14] study conducted among Polish teenagers, the less satisfied with their bodies more often consumed unhealthy energy-dense food (fatty and sweet food, i.e., pizza, chips, foods with dressings, chocolate bars or wafers, cakes) rather than healthy food (fruit and vegetables).

To conclude, it should be emphasized that adolescents dissatisfied with body weight did not demonstrate healthy eating behavior. An important argument pointing to the lack of healthy eating behavior is that even underweight students dissatisfied with their body weight promoted sweets and fats in their diet, but not health-promoting products. Sweet drinks, sweets and fast food are a source of empty calories, and their share in the total energy supply for children and adolescents is significant [45,46]. Consumption of sweetened beverages among adolescents is higher than in other age groups and is associated with a greater risk of weight gain, obesity and diet-related diseases [30]. HBSC data also showed gender an age differences in the daily consumption of sweets and sweetened beverages in adolescents (girls more often than boys consumed excessive amounts of sweets, and the percentage of adolescents drinking sweetened beverages more often than once a week was definitely higher among boys; the percentage of students consuming sweets and sweetened beverages increased with age) [5,6]. While limiting the consumption of high-energy products in groups of overweight young people should be considered justified, limiting the consumption of healthy products to regulate weight should be considered inappropriate [45,47,48]. Therefore, special attention should be paid to promoting the consumption of fruit and vegetables. In 10 European countries, most adolescents did not consume 400 g of fruit and vegetables daily, as recommended by WHO [49] or at the recommended frequency [6]. Nutrition education should also emphasize the differences in the calorific value of fruits and vegetables, promoting vegetables, as eating fresh fruit at least one portion a day was more likely declared by overweight and obese adolescents, and less likely by underweight students among those dissatisfied with their body weight, which should be considered inappropriate, as fruit can be a source of sugar in the diet. There is also evidence that weight regulation and an increase in body mass satisfaction can also be supported by the proper consumption of milk and dairy products [50,51] and wholegrain cereal products [52]. Fish consumption was associated with better school performance and cognitive testing in adolescents [53,54]. Therefore, it seems that nutrition education should be targeted at adolescents from older age groups who were more likely to have poor diet quality [6,55,56] and girls who were more likely to find their body weight inappropriate [6]. It is also extremely important to include a positive body image, proper body weight assessment and healthy lifestyle in health promotion and health education programs for youth.

### 4.3. Study Strengths and Limitations

To our knowledge, there are no nationwide studies on the effects of dissatisfaction with body weight on eating behavior conducted for all age categories of adolescents in Poland. Therefore, the important strengths of this research are the large and randomly selected sample of adolescents and anthropometric data related to their body size. The body weight status was assessed based on the data obtained through measurements, not self-reported data of body height and weight, which is a common procedure in large studies [3]. All anthropometric measurements and the eating behavior study were carried out by trained dietitians, based on a standardized procedure that ensured reliable results and minimized bias. Additionally, the questionnaire was validated prior to the main study and contained questions about the consumption of the main food groups.

Our study is subject to some limitations. The small number of questions with limited answer options (“yes” or “no”) in the questionnaire may be a limitation, also in the assessment of body weight dissatisfaction (“yes” or “no”). Dissatisfaction with body weight was a subjective experience of the subjects and was not assessed on the multi-level scale used in this type of study. It is possible that additional factors may affect body weight satisfaction. The questionnaire used in our study was validated for repeatability [15], but not in relation to another method. However, the questions have been developed on the basis of large, international studies on the nutritional behaviors of school-aged children [5,30], and include the most important healthy and unhealthy behaviors concerning nutrition. The cross-sectional nature of the study allowed us to identify the associations between socio-demographics and behavioral factors. However, we could not establish the causes and effects of the relationships between the analyzed factors.

## 5. Conclusions

This study provides new data on dissatisfaction with body weight and nutritional behavior of adolescents in two important aspects. First of all, the results of this study supported the existing evidence for a relationship between gender, age, body weight status, screen time and dissatisfaction with body weight in a large national random sample of Polish adolescents. Secondly, our results supported the relationship between dissatisfaction with body weight and unhealthy dietary behaviors among adolescents. These findings can help dietitians, nutritionists and healthcare professionals to provide age-specific and gender-specific nutrition strategies to promote healthy lifestyle among school-going adolescents. The interventions should be implemented at family, school and community levels.

## Figures and Tables

**Figure 1 nutrients-12-02658-f001:**
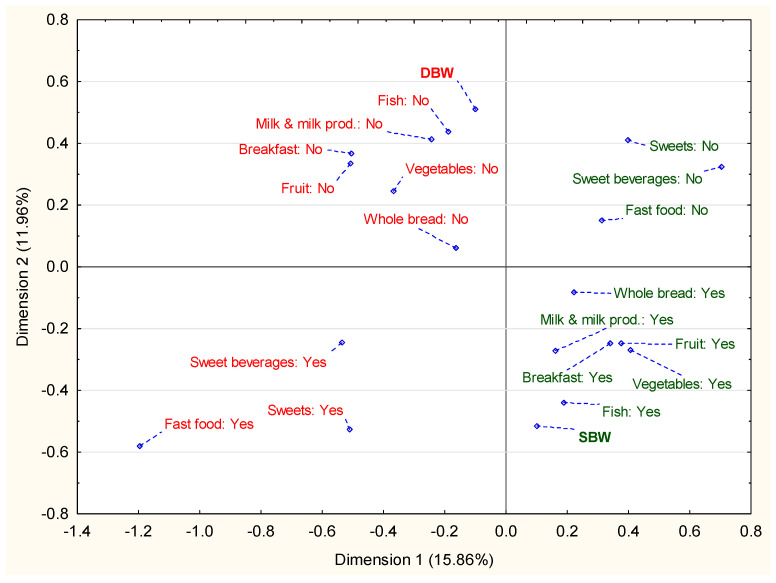
The results of the analysis of correspondence of eating behaviors in SBW (satisfied with body weight—green) or DBW (dissatisfied with body weight—red) groups (*n* = 14,044).

**Table 1 nutrients-12-02658-t001:** Characteristics of the study group (*n* = 14,044) depending on satisfaction (SBW) or dissatisfaction (DBW) with body weight and the odds ratio (OR) for DBW depending on analyzed factors.

Factor	Total [%]	SBW [%]	DBW [%]	*p* [Chi^2^]	OR for DBW [95% CI]
**Gender**					
girls	53.8	43.4	64.1	***	2.321 (2.169–2.484) ***
boys	46.2	56.6	35.9		0.431 (0.403–0.461) ***
**Age [years]**					
≤14	22.4	25.2	19.7	***	0.736 (0.680–0.797) ***
15	14.3	14.9	13.8		0.913 (0.831–1.004)
16	17	16.7	17.2		1.036 (0.948–1.131)
17	19.8	18.3	21.3		1.205 (1.109–1.309) ***
≥18	26.5	24.9	28		1.165 (1.081–1.256) ***
**Level of education**					
secondary	42.1	45.5	38.7	***	0.758 (0.709–0.811) ***
upper secondary	57.9	54.5	61.3		1.319 (1.233–1.411) ***
**Body weight status**					
underweight	5.1	5.2	4.9	***	0.943 (0.811–1.096)
normal	76.7	84	69.5		0.444 (0.399–0.469) ***
overweight	11.6	8.4	14.8		1.893 (1.701–2.107) ***
obese	6.6	2.4	10.8		4.983 (4.199–5.915) ***
**Screen time [h]**					
0–2	41.4	42.8	39.9	***	0.887 (0.829–0.948) ***
>2–4	42.5	42.6	42.5		0.997 (0.932–1.066)
>4	16.1	14.6	17.6		1.249 (1.141–1.368) ***

*** *p* ≤ 0.001—significant differences between SBW or DBW, the Pearson’s chi-square test or the Wald test (OR values).

**Table 2 nutrients-12-02658-t002:** Nutritional behaviors of the individuals (*n* = 14,044) depending on satisfaction (SBW) or dissatisfaction (DBW) with body weight and the logistic regression analyses for the association of SBW or DBW with nutritional behaviors.

Factor	SBW [%]	DBW [%]	OR for SBW [95% CI]	OR for DBW [95% CI]
**Having breakfast every day before leaving for school**	**65.8**	**53.9 *****	**1**	**0.607 (0.567–0.650) *****
underweight	4.9	5.2	0.807 (0.649–1.004)	1.120 (0.899–1.396)
normal	84.6	70.3	1	1
overweight	8.2	14	0.896 (0.751–1.070)	0.867 (0.759–0.992) *
obese	2.3	10.5	0.850 (0.618–1.169)	0.906 (0.778–1.056)
**Consuming fresh fruit every day (at least 1 serving)**	**58**	**57**	**1**	**0.960 (0.898–1.027)**
underweight	5	3.9	0.925 (0.747–1.146)	0.665 (0.534–0.828) ***
normal	83.9	67.6	1	1
overweight	8.6	16	1.069 (0.900–1.271)	1.298 (1.132–1.488) ***
obese	2.5	12.5	1.192 (0.868–1.638)	1.551 (1.322–1.820) ***
**Consuming vegetables every day (at least 2 servings)**	**49.1**	**46.0 *****	**1**	**0.886 (0.829–0.947) *****
underweight	5.4	4.6	1.075 (0.870–1.329)	0.921 (0.739–1.147)
normal	83.6	68.3	1	1
overweight	8.5	15.6	1.032 (0.871–1.223)	1.144 (1.001–1.307) *
obese	2.5	11.5	1.072 (0.787–1.459)	1.153 (0.990–1.343)
**Drinking milk or milk beverages every day**	**62.2**	**58.6 *****	**1**	**0.860 (0.803–0.920) *****
underweight	4.6	4.7	0.735 (0.594–0.910) **	0.876 (0.703–1.090)
normal	84.1	69.9	1	1
overweight	8.8	14.9	1.129 (0.945–1.349)	1.011 (0.883–1.158)
obese	2.5	10.5	1.159 (0.837–1.603)	0.932 (0.798–1.087)
**Consuming whole-grained bread every day**	**42.4**	**42.5**	**1**	**1.002 (0.937–1.071)**
underweight	5	4.5	0.951 (0.767–1.180)	0.902 (0.721–1.127)
normal	83.7	68.3	1	1
overweight	8.6	15.2	1.057 (0.891–1.254)	1.087 (0.950–1.244)
obese	2.7	12	1.241 (0.911–1.690)	1.253 (1.075–1.460) **
**Consuming fish at least once a week**	**52.6**	**47.5 *****	**1**	**0.816 (0.764–0.872) *****
underweight	4.5	4.7	0.755 (0.610–0.934) **	0.962 (0.773–1.197)
normal	83.8	68	1	1
overweight	9.1	15.4	1.187 (1.001–1.409) *	1.122 (0.982–1.283)
obese	2.6	11.9	1.182 (0.866–1.614)	1.258 (1.080–1.465) **
**Drinking sweet beverages few times a week**	**57.5**	**56.1**	**1**	**0.943 (0.882–1.008)**
underweight	4.8	6.1	0.847 (0.684–1.047)	1.759 (1.390–2.225) ***
normal	84.1	69.8	1	1
overweight	8.6	14	1.059 (0.891–1.258)	0.876 (0.766–1.002)
obese	2.5	10.1	1.117 (0.815–1.531)	0.846 (0.726–0.985) *
**Consuming sweets more than once a day**	**45.3**	**42.4 *****	**1**	**0.891 (0.834–0.953) *****
underweight	6.2	6.9	1.369 (1.106–1.693) **	1.826 (1.462–2.279) ***
normal	85.2	73.8	1	1
overweight	6.5	12.2	0.638 (0.534–0.761) ***	0.658 (0.572–0.756) ***
obese	2.1	7.1	0.798 (0.583–1.093)	0.472 (0.399–0.559) ***
**Consuming fast food more than 2 times a week**	**20.9**	**20.5**	**1**	**0.976 (0.899–1.059)**
underweight	5.6	7.3	1.122 (0.871–1.447)	1.498 (1.180–1.902) ***
normal	83.1	76	1	1
overweight	8.6	10.2	1.054 (0.857–1.296)	0.570 (0.473–0.686 ***
obese	2.7	6.5	1.221 (0.851–1.752)	0.484 (0.386–0.607 ***

* *p* ≤ 0.05; ** *p* ≤ 0.01; *** *p* ≤ 0.001—significant differences between SBW or DBW, the Pearson’s chi-square test or the Wald test (OR values). The bold data are the results of separate analyzes.

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
