# Peer review of "Dissatisfaction with Body Weight among Polish Adolescents Is Related to Unhealthy Dietary Behaviors"

_nutrients, 2020, doi:10.3390/nu12092658_

Round 1

Reviewer 1 Report

The authors provide a comprehensive overview of the large sample of adolescents regarding body weight and dietary behaviors in Poland. The authors could provide a few more details in order to improve the manuscript.

Abstract:

  1. It would be great to include one sentence in the beginning to provide a rational for the study, why is it important to provide data on socio-economic factors that affect weight dissatisfaction and dietary behaviors.

Introduction:

  1. I am missing an overview what the other studies (with smaller sample sizes) have found. Are there interesting associations that need validation? E.g. are there results one could doubt wherefore one would need more statistical power, or reliable results?
  2. What is the motivation to look for eating/weight related factors, e.g. what is the prevalence of eating disorders in Poland, are there concerning rises in incident rates wherefore it is more important to look are closely into associated constructs?

Method:

  1. Please remove the word “simple” page 3 line 111. Evaluative statements should not be in the method section.
  2. The answers for the nine questions for the eating behavior were also given as “yes” and “no”? Did the questions also include items on physical activity as indexed by the title of the pyramid? Will this data be published somewhere else?

Results:

  1. Please state more clearly; what different odds ratio entails (page 4 line 156). The DBW had more or less breakfast ? Had more or less milk every day?
  2. What do the authors mean by frequency (page 5 line 174), did the questions entail also answers on frequency rather than just yes/no?
  3. What do the authors mean by “which should be considered inappropriate” (Page 5, line 179). Evaluative statements should be moved to the discussion
  4. What do the authors mean by “…did not significantly improve frequency of consumption”…improve compared to when?who? The same with is the case with “modified” (line 183). It reads as there was some intervention, with before and after measurements…?
  5. Please delete “the” before body weight satisfaction (page 6 line 200)

Discussion:

  1. Generally I would suggest to move some of the details of the discussion to the introduction. For example Page 7 line 220-224, move this or some rephrased version to the introduction as I was missing the rational there as outlined above
  2. For the limitation section I would suggest to elaborate a bit more on the “limited possibilities of answers in the questionnaire” in regards to the fact that body dissatisfaction can be certainly displayed on a continuum that makes assessment of associated factors more easy than categorical data.
  3. Page 10 line 336/337 do the authors have any of how prospective studies could overcome this, maybe give a short outlook of next steps for future research

Reviewer 2 Report

In their article "Dissatisfaction with body weight among Polish adolescents is related to unhealthy dietary behaviors," the authors present interesting relationships between body weight dissatisfaction and age, gender, educational level, body weight status, and screen time. The large, representative sample of this study is unique for this area of research and can provide important information about who may be particularly at-risk for body dissatisfaction. However, the article would strengthened by addressing the following:

-In general, the introduction needs to focus more on body dissatisfaction specifically. It is not made clear how certain concepts (e.g. WHO health promotion campaign mentioned on line 51, food advertising mentioned on lines 45-48) relate to body dissatisfaction. This is done well in the conclusion; it is stated that body dissatisfaction may be associated with and/or lead to other unhealthy nutrition behaviors. However, this is not clearly outlined in the introduction and it does not make it clear to the reader what the article will be focused on. The relationships between body dissatisfaction and psychophysical health should be clearly stated.

-In addition to that, the introduction does not highlight why this study is novel and where the gaps in research are- is it that this study has not been done in Polish adolescents specifically, is it that this study included several socio-demographic measures, etc? Has a similar study been conducted in another country, and why would you hypothesize that Polish adolescents would differ? Clarifying the gaps in the research and how this study fills these gaps will highlight why it is novel.

-The sentence beginning on line 65 (This study was undertaken…) implies that these analyses were the primary aim of the study. However, in the next paragraph, it is stated that these measurements were part of a larger program. If these analyses were secondary to the primary aim of the study/program, that should be clearly stated. It is currently unclear if these measures were taken as part of the program and a secondary analysis conducted, or if schools were selected from this program for a prospective analysis of body dissatisfaction.

-Line 86: The sentence “All parents and students over 16 years of age were asked to give their written consent to participate in the study” is confusing. Did parents consent for children under 16, and children over 16 consented for themselves? Needs clarification.

-Line 103: It is not clear what “type of school” means.

-Most importantly, more details about the measure of body dissatisfaction are needed. Beginning at line 104, it is stated that the question “Are you satisfied with your body weight?” was used to assess body satisfaction/dissatisfaction. Has this way of measuring body satisfaction/dissatisfaction been published or otherwise validated? There are other measures of body weight satisfaction/dissatisfaction (most commonly a body image scale- see citations below). Additionally, framing the question as being satisfied or dissatisfied with body weight could be interpreted very differently for different children; some may be focusing on the number on the scale while others would interpret that as body size or shape. It is imperative that the authors are able to justify the use of this question as opposed to other, validated measures. This should also be addressed in limitations section.

Candy CM, Fee VE. Underlying dimensions and psychometric properties of the eating behaviours and body image test for preadolescent girls. Journal of Clinical Psychology 1998;27:117–127.

Cohn LD, Adler NE, Irwin CE, Millstein SG, Kegeles SM, Stone G. Body -figure preferences in male and female adolescents. Journal of Abnormal Psychology 1987;96:276–279.

Sherman D, Iacono W, Donnelly J. Development and validation of a body rating scale for adolescent females. International Journal of Eating  Disorders 1995;18(4):327–333.

Thompson JK, Altabe MN. Psychometric qualities of the Figure Rating Scale. International Journal of Eating Disorders 1991;10:615–619.

-It is not clear when each measurement (body satisfaction/dissatisfaction, anthropometrics, nutrition questionnaire) was taken. Were these all measured on the same day? The timeline is very unclear, and it is not clear until line 107 that these measures were taken before some sort of intervention/program (baseline). Perhaps a figure outlining the study procedures and timeline would be helpful to the reader.

-Line 106: It is unclear what “qualified” means. Does this mean that dietitians are registered or otherwise certified, does this describe their training to administer this task, etc?

-Lines 107 and 111: Both refer to “the questionnaire.” Are these the same questionnaire? Each questionnaire should be named for clarity.

-Beginning at line 111, it is unclear how these questions were asked. Were they simple yes/no questions, did they assess frequency (days/week of each behavior), etc?

-Line 119: The sentence states that “The nutritional behaviors of the respondents were analyzed based on the division into SBW and DBW groups.” Was this questionnaire scored for one total score, or was each of the 9 questions analyzed separately? This becomes clear in the results section but should be clarified here as well.

-Although a Chi-square test is appropriate, the analyses would be strengthened by choosing models in which control variables can be incorporated. Given the size of this sample, it would be ideal (and would add more to the literature) to examine effects of socio-demographic factors controlling for other confounders. For example, the authors found sex differences in DBW. Sex should then be controlled for when analyzing dietary behavior data, as much evidence suggests sex differences in other aspects of eating. Additionally, on line 234, the authors state that “As documented in earlier study [17], the average screen time increased with age, which can also cause an increase in dissatisfaction with body weight.” Therefore, as an example, age should be controlled for when analyzing body dissatisfaction and screen time associations. Known or hypothesized confounders should be controlled for when possible. However, if choosing to keep the Chi-square analyses, it should be specified if these analyses were conducted as chi-square tests of independence or chi-square goodness-of-fit tests.

-Line 150: Use of the word “impact” suggests causality- consider rephrasing to reflect that they are associated since causality cannot be determined.

-Line 179: It is unclear what “which should be considered inappropriate” means.

-Line 180: It is not clear what “abnormal” body weight is. If this is referring to children with underweight and/or overweight/obesity, it should not be phrased as “abnormal” since this is confusing (and potentially stigmatizing).

-Lines 181-185: Wording such as “improved,” “modified,” and “maintained” suggests an effect of time. However, my understanding is that this was assessed at one time point. Wording should be changed to reflect cross-sectional nature of the study.

-Line 211: Use of the word “affect” implies causality, which cannot be established (other counfounders may be present). Should use “associated with” instead.

-Line 292: Authors state that “it should be emphasized that adolescents dissatisfied with body weight lacked healthy nutrition.” This is overstating results, as it is not necessarily true that they lacked healthy nutrition. Rather, they did not demonstrate as healthy of nutrition behaviors as a comparator group.

-Line 294: The use of the word “modified” indicates causality and change over time. However, this was a cross sectional analysis and these behaviors were only associated with underweight.

-Lines 342 and 344: Should not use the word “confirmed.” Rather, state that this supports existing evidence.

-Sex and gender are used interchangeably in this article; however sex is biological and gender is identity. Need to choose which one was measured and keep same word throughout.

-The conclusion is strong and ties together current results with previous published findings.

Round 2

Reviewer 2 Report

The article "Dissatisfaction with body weight among Polish adolescents is related to unhealthy dietary behaviors" contributes additional evidence that there are individual differences in body dissatisfaction in a large, representative sample. However, a couple more revisions will strengthen the article:

-It is still unclear why the methodology for collection of body weight dissatisfaction was chosen, especially when the "standard" way to collect this is fairly quick and simple (as described in the publications cited in previous review). More justification is needed, as this is the central topic of the article.

-Rather than refer to children with underweight & overweight /obesity as having "abnormal" or "incorrect" body weight, the weight groups themselves should be used to describe the groups and people-first language should be used (e.g. starting at line 270, the sentence should read "Thus, our results indicate a strong need for the increase in educational activities promoting regular consumption of breakfast, especially among adolescents with underweight or overweight/obesity, or with body weight dissatisfaction.")

Thank you for your work on this research and this article.
